# Research the Thermal Decomposition Processes of Copolymers Based on Polypropyleneglycolfumaratephthalate with Acrylic Acid

**DOI:** 10.3390/polym15071725

**Published:** 2023-03-30

**Authors:** Akmaral Zh. Sarsenbekova, Gaziza M. Zhumanazarova, Yerkeblan M. Tazhbayev, Gulshakhar K. Kudaibergen, Saule K. Kabieva, Zhannur A. Issina, Aigul K. Kaldybayeva, Arailym O. Mukabylova, Muslim A. Kilybay

**Affiliations:** 1Chemistry Faculty, Karaganda Buketov University, Karaganda 100024, Kazakhstan; 2Department of Chemical Technology and Ecology, Karaganda Industrial University, Temirtau 101400, Kazakhstan; 3National Center for Biotechnology, Astana 100000, Kazakhstan; 4Department of Pharmaceutical and Toxicological Chemistry, Pharmacognosy and Botany, Asfendiyarov Kazakh National Medical University, Almaty 050060, Kazakhstan

**Keywords:** dynamic thermogravimetry, polypropyleneglycolfumaratephthalate, acrylic acid, kinetic analysis, activation energy, preexponential factor, thermal degradation

## Abstract

Kinetics of thermal degradation of polymeric materials is usually studied by weight loss at a constant temperature or during heating. Hence, the activation energy and other kinetic parameters of the thermal destruction process are determined. One of the fastest and most accessible methods for studying the kinetics of these processes is TGA. Weight methods of TGA do not provide an opportunity to judge the proportion of gaseous degradation products. This is especially true for processes associated with the release of hydrogen and other substances with low molecular weights, the accuracy of determining the amount of which by the weight method is low. Meanwhile, the study of the gas evolution process can provide additional information about the kinetics and mechanism of thermal destruction processes. Of great interest is also the joint study of the total weight loss and gas evolution during the polymer heating. Using mass spectrometry, IR spectroscopy combined with thermal analysis (TGA/DSC-IR and TGA/DSC-MS) we have defined product composition and thermal destruction kinetics. As a result of the TGA/DSC-MS study of gaseous products of thermolysis in nitrogen atmosphere, there were found products with 44, 45, 59, 60, 68, and 88 phr. Quite a similar pattern for p-PGFPh:AA copolymers is also observed in TGA/DSC-IR studies: the same products and the same temperature range. However, in contrast to the TGA/DSC-MS study, CO release was also recorded by this method (weak signal). Kinetic characteristics of the processes were determined based on Friedman, Ozawa-Flynn-Wall and modified NPC methods. Obtained values of the activation energy and thermodynamic characteristics make it possible to predict the composition of polymers, which make a significant contribution to the development of theoretical ideas about the features of the physicochemical properties of polymers.

## 1. Introduction

The most prioritized research for making a new generation of polymers is the one aimed at obtaining polyester resins and their copolymers with the required properties. The selection of polyester resins largely determines the technology for the production of composite materials and significantly affects the formation of the properties of the resulting products [1,2,3,4,5]. The range of production areas in which the use of polyester resins plays a leading role is quite large and it includes the automotive industry, the production of modern building materials, etc.

Unsaturated polyester resins (including polypropyleneglycolfumaratephthalate) form the basis of decorative finishing materials, structural products, as well as hydroponics, known as «artificial soil» and moisture sorbents that are used in the cultivation of crops [6]. Cured polyester resins are structural materials with high strength, hardness, wear resistance, excellent dielectric properties, high chemical resistance, and environmental safety during operation. Due to the relatively low molecular weight of unsaturated polyesters, their solutions have a low viscosity, which allows them to be processed without applying high pressures and, often, without complex technological equipment. One of the advantages of polyester resins is their low cost, which is two times lower than the cost of epoxy resins. There is an apparent explanation for that, their factor contributes to a significant simplification and reduction in the cost of the work process, an increase in product quality, and production without harming the health of employees and the environment [7,8]. These properties are primarily expressed in polyester resins, thereby they appear in polycondensation polymers quite clearly [9,10,11,12]. All this has led to active growth in their use and to the emergence of a wide range of fundamentally new methods of production, processing, and use [13]. With the variety of their positive properties, polyester resins have a number of disadvantages. First, they are prone to aging, and second, polymers are combustible materials, which prevents their wide distribution [14,15]. Despite the predominantly long-term use of polymer resin materials, with age, they lose their initial physicochemical properties and begin to be exposed to various harmful external influences. All these make them unsuitable for their intended use. Therefore, it is very important to study the processes of kinetic and thermodynamic nature, and factors that allow controlling the process of material degradation, rate, and characteristics of its aging.

Several scientific approaches are considered recognized methods for determining the nature and characteristics of activation energy in science. One of the most relevant is the method patented under No. 2216012 [16]. This method is based on the fact that molecular mobility in polymers is directly related to thermal degradation since polymer molecules due to bond breaking from thermal degradation have greater mobility. In turn, molecular mobility is inversely proportional to the molecular relaxation time, so by studying this last characteristic, one can judge the processes of molecular mobility and thermal degradation. This method has only one drawback: it is applicable only for one class of materials, polymer electrets. Of great interest is the method developed and patented under RU No. 2554623 [17]. This method covers a specific area associated with environmental restrictions on the use of appropriate polymeric materials, which leads to the need for careful studies of gaseous products of thermal degradation of materials. The test time here is chosen based on the method of accelerated tests for climatic aging SS (State Standard) 9.707-81. Therefore, implementation of the last analogous method is also associated with a rather significant investment of time and resources, without considering the cost of conducting gas chromatography–mass spectrometric studies. The procedure according to the article of SS 9.715-86 is also interesting to study (Common System of Protection from Corrosion and Aging. Polymeric Materials. Test Methods for Resistance to Temperature). The material sample is heated at a given rate in the air or a medium of a given composition, continuous recording on the thermoanalytical curve of the processes occurring in the material, and determining the resistance of materials to the effects of temperature using one or more of the following indicators characterizing these processes: conditional temperatures of start (*T_s_*) and termination (*T_t_*) of the processes; the change of mass of the sample (Δ*m*); the constant of process rate (*K*)’; and activation energy (*E*) which characterizes the dependence of the rate of change of the mass of the sample from temperature. The value (E) is determined in the prototype according to the following sequence of actions. The advantage of the considered method over the traditional method of accelerated aging lies in its rapidity.

Compared to other classical methods of materials aging, the rapidity of the considered method determines its speed and ease against the background of others. However, it should not be used in all cases, as there is a decrease in the accuracy of the analysis of kinetic parameters, for this reason, it is used for a possible number of areas of additional predictive tools.

Technology for producing one of the main sources of raw materials, polypropyleneglycolfumaratephthalate, which is necessary for producing the proposed moisture sorbent, has been fully developed [18]. At the same time, before our research, the process of studying the thermal stability of copolymers based on unsaturated polyester resins had not yet been studied by specialists.

The aim of this work is to study the kinetics and mechanism of the primary stages of thermal decomposition of copolymers based on polypropyleneglycolfumaratephthalate with acrylic acid using thermal analysis using infrared and mass spectroscopy, including quantitative determination of thermal degradation products composition and definition of this processes primary stages’ kinetic parameters at different heating rates.

## 2. Materials and Methods

### 2.1. Materials

Used materials and detailed methods are presented in the Appendix A.

### 2.2. Synthesis of p-PGFPh

Polypropyleneglycolfumaratephthalate was obtained by the polycondensation reaction of propylene glycol, phthalic anhydride, and fumaric acid at a temperature of 453–473 K. The polycondensation was carried out according to the standard procedure [10] in the presence of an aluminum chloride catalyst under nitrogen in order to avoid undesirable gelatinization processes. Polycondensation was carried out for 16 h.

### 2.3. Synthesis of Copolymers of p-PGFPh

The radical copolymerization of p-PGFPh with AA was carried out in a dioxane solution (1:1 by mass) in the presence of benzoyl peroxide (BP) as an initiating agent at 333 K. The ampoules were purged with an inert gas (N_2_) for 30 min. Synthesis was carried out for 52 h. The resulting copolymer was washed with dioxane and dried to a constant weight in a vacuum oven.

### 2.4. Methods

The study of the kinetics of the thermal degeneration process is performed using TGA-DSC-IR, TGA-DSC-MS thermal analysis with the curve registration on Netzsch Jupiter STA 449 F3 combined with IR, MS analyzer Bruker FTIR alpha II. The sample weight is 30 mg in powder form. The TG curves of the samples were taken in the temperature range of 50–600 °C and heating rates of 5, 7.5, 10.0, and 12.5 °C/min in an argon or nitrogen atmosphere.

The experimental data is processed by the program “Origin Pro 8.1”, “Anaconda (Python distributions with Numpy, Matplotlib, and Scipy packages for data processing and visualization)”.

The experimental data is processed by the program «Origin Pro 8.1», «Anaconda (Python distributions with Numpy, Matplotlib, and Scipy packages for data processing and visualization)».

## 3. Results and Discussion

Previously, we synthesized copolymers with high moisture-absorbing properties by the reaction of radical copolymerization of unsaturated polyester resins [15], which can be used as hydrogels in the processing of vegetable crops. It is shown that copolymerization proceeds according to the following scheme (Figure 1):

Bringing moisture sorbents into widespread use for the purpose of growing crops in agricultural production raises the problem of expanding the use of hydrogels. During the long-term use of artificial soil created on the basis of hydrogels, we will still face the process of destruction: aging and disturbance of their structure. Undoubtedly, this process will lead to a clear decrease in crop yields. At the same time, the problem of identifying substances that, under the influence of various natural and technogenic factors, decompose the structure of polymer gels added to natural soil, as well as the structures that form the basis of artificial soil, remains unexplored. For this reason, the study of the thermal stability of the obtained moisture sorbents and the established basic patterns of the degradation process are needed in the study.

The composition of the products and kinetics of thermal decomposition of p-PGFPh:AA copolymers were studied by thermal analysis using infrared and mass spectroscopy (TGA/DSC-IR and TGA/DSC-MS). Its advantage over other research methods is the ability to determine the kinetic parameters of the decomposition of condensed substances with the simultaneous identification of the resulting gaseous products.

In the course of our study, we were able to determine the main kinetic parameters of the decomposition of p-PGFPh:AA (6.77:93.23), p-PGFPh:AA (86.67:13.33) copolymers using isoconversion models. The results of the research showed the practical value of this technique. The resulting thermogravimetric analysis curves and decomposition rates are shown in Figure 2.

As shown in Figure 2a, p-PGFPh:AA (6.77:93.23) copolymer sample in nitrogen starts decomposing at ~100 °C. Then up to ~150 °C minor sample decomposition with highly volatile substances yield can be seen. The main stage of p-PGFPh:AA (6.77:93.23) copolymer thermal decomposition occurs between ~250 °C and ~450 °C. After this sample, mass stabilization can be seen. Mass loss speed (Figure 2b) at a heating rate increase changes between 5 °C/min and 12.5 °C/min. Charts clearly show curve shifts to a higher temperature area, and this is shown by the mass loss peak shape (Figure 2b) of p-PGFPh:AA (86.67:13.33) copolymer. They have symmetrical sides which suggest a complex copolymer decomposition mechanism; its main stages occur due to ester link disruption, which is unstable at 300 °C. Some volatile substances can be determined by a specific strong peak of IR radiation absorption, where pyrolysis products in gas form are mainly CO_2_ (2300 cm^−1^) and C=O (1720 cm^−1^). This conclusion is also proven by IR spectrometric analysis results. In the IR spectrum (Figure 3a) of p-PGFPh:AA (6.77:93.23) copolymer decomposition products, there is a wide line with a maximum at 1719.11 cm^−1^ which belongs to valence vibrations of the C=O carbonyl groups. There is also an absorption band 2944.55 cm^−1^ of valence vibrations of C–H linkages in CH- and CH_2_-groups as well as absorption bands with 1450, 1155, 1060 cm^−1^ of deformation vibrations of C–H linkages in C–O and C–O–C groups. p-PGFPh:AA (86.67:13.33) copolymer thermal decomposition occurs similarly, except that in the recorded IR spectrum of p-PGFPh:AA (86.67:13.33) copolymer decomposition at temperatures between 400 and 600 °C CO_2_ groups (2360 cm^−1^), valence vibrations line intensity increases. Thermograms and IR spectra are given in the Appendix A.

The thermogram of p-PGFPh:AA copolymers at ratios M_1_:M_2_: (6.77:93.23) (Figure 4a) and (86.67:13.33) (Figure 4b) shows thermal effects belonging to both p-PGFPh and AA. In contrast to AA (Figure 4) and their equimolar mixtures with p-PGFPh, the thermal effects of AA in copolymers are shifted to the high-temperature area, while the melting points of AA in the composition of copolymers also increase. These results are consistent with the direct determination of the melting points of the p-PGFPh monomer and copolymers (Table 1, Figure 4), as well as with the results of the IR spectrometric study of the decomposition products of p-PGFPh:AA copolymers upon heating: the destruction of the inclusion compounds begins with the removal from the cavity, first water molecules, then acrylic acid, and only after that begins the thermal destruction of polypropyleneglycolfumaratephthalate.

As can be seen from the results of differential thermal analysis (Table 1, Figure 4), for p-PGFPh:AA copolymers (6.77:93.23) ΔC_p_ = 26.50 J/g °C, then for the p-PGFPh:AA copolymers (86.67:13.33)—ΔC_p_ = 25.37 J/g °C. The thermograms undergo the most significant changes in the zone of the peak melting of the crystalline phase. If for the copolymer p-PGFPh:AA (6.77:93.23) there is a wide asymmetric melting peak of ΔT_max_ = 340 °C (the value of ∆H*_f_* was 68.52 J/g), then for the copolymer p-PGFPh:AA (86.67:13.33) ΔT_max_ = 350 °C (the value of ∆H*_f_* was 100.98 J/g). These results allow us to conclude that the size dispersion of the crystallites, which mainly affects the width of the melting peak, is approximately equal for all copolymers. This is evidenced by the proximity of the melting thermograms p-PGFPh: AA (6.77:93.23) and p-PGFPh:AA (86.67:13.33).

Another good method for obtaining information about the chemical structure of organic substances is the use of mass spectrometry. A combination of TGA/DSC with a mass spectrometer (MS) is becoming increasingly popular due to its ability to detect very small amounts of impurities.

Figure 5 shows that the chromatogram contains a small number of peaks that correspond to different fragments and different lengths of p-PGFPh:AA polymer chains. Obviously, the shortest retention time will correspond to lighter components. In our case, the «lightest» fragments of the p-PGFPh:AA copolymers will be desorbed at the very beginning of the heating process. Figure 5a,b shows the mass spectra of the products of thermal decomposition of p-PGFPh:AA copolymers at different ratios (6.77:93.23 mol.% and 86.67:13.33 mol.%). At T_max_ = 340 °C, an intense peak (M/Z = 44) is observed in the mass spectrum (Figure 5a) for p-PGFPh:AA copolymers (6.77:93.23) and T_max_ = 350 °C for p-PGFPh:AA (86.67:13.33). In the same temperature range, there is a peak M/Z = 45 in smaller amounts. On these mass spectra (Figure 5), the most intense peaks correspond to a molecular ion with a mass of 44 a.m.u. Search results in the NIST mass spectrum database indicate that this peak corresponds to carbon dioxide (CO_2_) (Figure 5). The release of a small amount of CO_2_ also occurs at ~538 °C for p-PGFPh:AA copolymers (6.77:93.23 mol.%) and ~550 °C for p-PGFPh:AA copolymers (86.67:13.33).

When comparing the samples’ thermograms, it can be concluded that the p-PGFPh:AA copolymers are thermally stable. Comparative analysis of the IR spectra of p-PGFPh:AA copolymers of various compositions heated to 800 °C in a nitrogen atmosphere showed that as the temperature rises, the intensity of the bands gradually decreases at 1155 cm^−1^, which characterize C–O groups, and then it gradually decreases (Figure 3), which indicates the destruction mainly by ether bonds of polypropyleneglycolfumaratephthalate. The process of destruction of the polymer by the ester bond, as well as the release of carbon dioxide during thermal destruction, can be represented by the following (Figure 1):

The analysis of the thermal destruction of gaseous products shows that, as a result of the thermal decomposition of copolymers p-PGFPh:AA (T_max_ = 340−350 °C), such toxic gaseous products as carbon monoxide CO and carbon dioxide CO_2_ are formed. According to TGA/DSC-MS analysis, acrylate units share an increase in p-PGFPh:AA copolymer (6.77:93.23 mol.%) that leads to twice the lower formation of toxic gaseous products, such as carbon monoxide CO and carbon dioxide CO_2_ (see Figure 5).

An objective analysis of p-PGFPh:AA copolymers thermal degradation process at components ratio 6.77:93.23 and 86.67:13.33 is possible in the case of process activation energies comparison since activation energy is the only reliable criterion that can be used for direct comparison. Kinetic analysis was performed considering ICTAC-2000 [19,20,21,22,23] protocol.

Kinetic parameters calculations on the TG curve are based on the formal kinetic equation:(1)$$\frac{d\alpha }{\mathrm{dt}}=\beta \frac{d\alpha }{\mathrm{dT}}=k\left(T\right)f\left(\alpha \right)=A\exp\left(-\frac{E_{\alpha }}{RT}\right)f\left(\alpha \right)$$
where α—stage of process completion, t—time, T—temperature, *β*—heating rate (K/min), *f*(α)—kinetic model, *k*(T)—speed constant depending on temperature according to Arrhenius equation with parameters A (pre-exponential factor), and *E_a_—*(activation energy), *R*—universal gas constant.

The differential kinetic equation for isothermal conditions appears as follows:(2)$$\frac{d\alpha }{\mathrm{dt}}=k\left(T\right)\cdot f\left(\alpha_{i}\right)$$
where α*_i_*—degree of reaction completion by the time t*_i_*. Conversion and integration of Equation (2) allows switching to the integral form of a kinetic equation:(3)$$\int_{0}^{\alpha_{i}}\frac{d\alpha }{f\left(\alpha_{i}\right)}=k\left(T\right)\cdot \int_{0}^{t_{i}}\mathrm{dt}\;\mathrm{or}\;g\left(\alpha_{i}\right)=k\left(T\right)\cdot t_{i}$$

For a non-isothermal experiment carried out at a constant heating rate $\beta =\frac{\mathrm{dT}}{\mathrm{dt}}=\mathrm{const}$, the transformation of Equations (2) and (3) can be represented as: (4)$$g\left(\alpha_{i}\right)=\frac{A}{\beta }\int_{{T-T}_{0}}^{T_{i}}\exp\left(-\frac{E}{\mathrm{RT}}\right)\mathrm{dT}$$
where T_0_—temperature corresponding to condition α*_i_* = 0.

The solution of integral of the right side of Equation (4) can be found only in an approximate form by expanding in a series:(5)$$g\left(\alpha_{i}\right)=\frac{AE}{\beta R}\left[\exp\left(-\frac{E}{{\mathrm{RT}}_{i}}\right)\right]\sum_{i=1}^{\infty }{\left(-1\right)}^{i+1}i!{\left(\frac{{\mathrm{RT}}_{i}}{E}\right)}^{i+1}$$

This solution, known as a primary analytical equation (PAE) [24], can be limited in terms of the number of terms in the series, if thermal energy defined by RT product is much less than E. If *i* = 1 condition is met, integral form of kinetic equation simplifies:(6)gαi=ARTi2βRexp−ERTi

In the case of complex processes involving several series-parallel stages, it is recommended to use isoconversion methods such as the Friedman (FR) isoconversion differential method [25,26] and Ozawa-Flynn-Wall (OFW) isoconversion integral method [27,28].

Isoconversion methods make it possible to obtain the dependences of the apparent activation energy on the depth of the process, regardless of the mechanism of the process and the function *f*(α) that characterizes it.

When using the Friedman method, the equation is presented in the form: (7)$$\ln\left(\beta \frac{d\alpha }{{\mathrm{dT}}_{\alpha }}\right)=\ln\left\{A_{\alpha }f\left(\alpha \right)\right\}-\frac{E_{\alpha }}{RT}$$
where A and E_α_ are kinetic parameters, the pre-exponential factor, and the activation energy, respectively; *R* is the universal gas constant; and T is the absolute temperature at time t. Here, α—stage of process completion, T—temperature, *β*—heating rate (K/min), *f*(α)—kinetic model, dα/dT_α_—the rate of conversion.

Since *f*(α) is a constant value at any fixed α, dα/dt conversion rate logarithm dependence for each heating rate *β* on 1/T is a straight line, its slope ratio is E_a_/R, and segment cut by this line on Y-axis equals ln{A_α_*f*(α)}. Preexponential factor estimation is generally performed based on the Arrhenius equation in first-order reaction assumption *f*(*α*) = (1 *− α*) averaging all dynamic heating rates (see. Figure 6).

The Ozawa-Flynn-Wall method uses an integral equation of the following form:(8)$$\log\beta =-1.052\frac{E_{\alpha }}{{\mathrm{RT}}_{\alpha }}5.3305+\ln\left\{\frac{R}{{\mathrm{AE}}_{\alpha }}\int_{0}^{1}\frac{d\alpha }{f\left(\alpha \right)}\right\}$$
where *f*(α) is the integral conversion function.

For data obtained in series of measurements taken at different heating rates *β* at fixed conversion rate α according to equation ln(*β)* dependence on 1/T will be a straight line with a slope ratio equal to −1.052 E_a_/R.

Analysis of obtained data in the Arrhenius coordinates gives parameters of thermal decomposition kinetic equations $k=A\cdot {exp}^{-E_{a}/RT}$ which are shown in Table 1, Figure 6, and in the Appendix A. For known α at the selected heating rates, the plot ln (β·dα/dT_α_) vs. (1000/T) was linear. At graphic representation of dependence (Equation (2)) for different rates of reaction flow (see Figure 6a,b), average values of activation energy corresponding to different stages of the process can be obtained. The values of the apparent activation energy (E_a_) for the two samples were obtained (Table 2) by evaluating the slopes of these straight lines (see Figure 6a,b).

Table 2 shows that polymer p-PGFPh ${\bar{E}}_{\mathrm{FR}}$ = 237.62 kJ mol ^−1^ has the highest average activation energy value. Thus, we can conclude that p-PGFPh requires more energy to break the ether bond. Identical activation energy values for p-PGFPh:AA copolymers at ratios 44.17:55.17 and 86.67:13.33 can be explained by the fact that their formation occurs through the same intermolecular reactions (Table 2). Activation energy of copolymer p-PGFPh:AA (86.67:13.33) is significantly lower than those of p-PGFPh, which suggests weaker intermolecular bonds.

To continue the study and find apparent activation energy, the non-parametric kinetics method was applied. Mathematical processing of thermogravimetric curves shown at Figure 2 allows for finding apparent activation energy using non-parametric kinetics method based on Equation (1)
(9)$$r=k\left(T\right)\cdot f\left(\alpha \right)$$

In continuation of the work, in order to determine the apparent activation energy, the method of nonparametric kinetics was applied. Mathematical processing of the thermogravimetric curves shown in Figure 2 makes it possible to determine the magnitude of the apparent activation energy using the nonparametric kinetics method based on Equation (1).

This method allows for calculating all kinetic parameters of a one-step process on the basis of one differential thermogravimetric curve (Figure 2a,b). The advantage of this method is that special assumptions about the size of reaction order and kinetic equation function matching are not needed.

Authors of studies [29] rely on the assumption that in the NPK method reaction, speed can be expressed in form of a matrix
(10)$$M=\left\{m_{ij}\right\}=\left\{k\left(T_{i}\right)\cdot f\left(\alpha_{j}\right)\right\}$$

The most important feature of that method is the fact that NPK method uses the algorithm of singular value decomposition (SVD) [30,31].
(11)$${M=USV}^{T}$$

Given matrix elements U and V are defined by the following expressions:(12)$$f\left(\alpha \right)=\left[u_{1},u_{2},\ldots u_{i}\right]\;\mathrm{and}\;k(T)=\left[w_{1}v_{1},w_{1}v_{2},\ldots w_{1}v_{1}\right]$$

Reaction speed values at different heating rates obtained this way were approximated using the NPK method and interpolated as a surface in 3D space (β⋅dα/dt, α, T) (Figure 7). This surface is organized as i×j matrix where lines correspond to different conversion rates from $\alpha_{1}$ to $\alpha_{i}$, and columns correspond to different temperatures from $T_{1}$ to $T_{j}$.

One of equations properly describing reactions characterized by presence of noticeable induction period followed by rapid reaction speed increase is generalized Šesták–Berggren kinetic equation [32]. $f\left(\alpha_{i}\right)=\alpha_{i}^{m}{\left(1-\alpha_{i}\right)}^{n}\left[{-\ln\left(1-\alpha_{i}\right)}^{p}\right]$, also known as *Prout–*Tompkins** (P–T) equation [33]
(13)$$f\left(\alpha_{i}\right)=\alpha_{i}^{m}{\left(1-\alpha_{i}\right)}^{n}\left[{-\ln\left(1-\alpha_{i}\right)}^{p}\right]$$

It was believed [34,35] that the thix kinetic equation, containing as many as three exponential terms, is able to describe any thermal analysis curves. Further mathematical analysis thermal analysis of Equation (8) has shown, however, that no more than two kinetic exponents are necessary. Therefore, after eliminating the third exponential term in Equation (13) the final form obtained is
(14)$$f\left(\alpha_{i}\right)=\alpha_{i}^{m}{\left(1-\alpha_{i}\right)}^{n}$$

Exponent’s m and n are taken as kinetic parameters, which describe the shape of measured thermal analysis curves. The Šesták–Berggren (Š–B) model, Equation (13) can also be understood in terms of the $f\left(\alpha_{i}\right)$, where $\alpha_{i}$($\alpha_{i}$) can bear the form of either function, $\alpha_{i}^{m}$, ${\left(1-\alpha_{i}\right)}^{1-n}$ and/or ${\left[-\ln\left(1-\alpha_{i}\right)\right]}^{p}$.

In the opinion of one of the authors [33], this equation should be considered as empirical; however, multiple ways of kinetic description based on using polynomial approximation while calculating via primary analytical equation are reduced to this relatively simple form. Moreover, as m values are within 1 ≥ m ≥ 0 [36] and limitation of the sum m + n = 2 [37] is possible, Equation (14) can be converted into the following integral form:(15)$$g\left(\alpha_{i}\right)=\frac{1}{n-1}{\left(\frac{\alpha_{i}}{1-\alpha_{i}}\right)}^{n-1}=k\left(T\right)\cdot t_{i}$$

Definition of the rough value n can be done using procedure recommended in the study [36]. Thus, Equation (14) factors ratio should comply with the following equation:(16)$$\frac{m}{n}=\frac{\alpha_{\max}}{1-\alpha_{\max}}$$
where α_max_—degree of reaction completion corresponding to maximum reaction speed.

As m + n = 2, value of n can be found from the simple correlation n = 2(1 − α_max_). Function selection was done by choosing the most suitable model in dα/d*t—*α coordinates at different heating rates *β*. $f\left(\alpha_{i}\right)=\alpha_{i}^{m}{\left(1-\alpha_{i}\right)}^{n}$ was used as *f*(α) function. Parameters *m* and *n* influence shape and peak position of curve dα/d*t* (Figure 8). Kinetic analysis results are shown in Figure 8 and Table 3.

For p-PGFPh:AA (6.77:93.23) copolymers (Table 3, Figure 8), the most suitable function is *f*(α) = α^0.60^(1 − α)^1.40^, and for p-PGFPh:AA (86.67:13.33) copolymers it is *f*(α) = α^0.52^(1 − α)^1.48^. As Figure 8a suggests, at the beginning, speed quickly increases due to self-acceleration, reaching maximum value at *α*_max_ = 0.60, and then it quickly decreases until almost complete decomposition stops. p-PGFPh:AA (86.67:13.33) copolymers decomposition process also shows self-acceleration until α = 0.5. After that, speed gradually decreases.

As Figure 9 suggests, activation energy values at different conversion rates for both copolymers have similar trends. At the beginning of the thermal degradation reaction, activation energy values are high, which is likely to be related to ester bonds disruption and carbon oxide emission. Then, there is gradual decrease of activation energy spent on thermal degradation of compounds modeling elements of primary chain and intermediate destruction products. At α > 0.5, a minor increase can be seen, which can be considered as thermal conversion of initial polymer with formation of new, partially linked structures. For (p-PGFPh:AA 6.77:93.23 mol.%), sample E_a_ rapidly rises at α = 0.6 and reaches its maximum of 245.45 kJ/mol at α = 1. For (p-PGFPh:AA 86.67:13.33 mol.%), sample activation energy E_a_ decreases at α > 0.5. This suggests low impact of ash content on bound carbon burning. Such an important kinetic parameter as average activation energy was defined as average E_a_ value. Average copolymer activation energy calculated using NPK was 204.01 kJ/mol (p-PGFPh:AA 6.77:93.23 mol.%) and 198.82 kJ/mol (p-PGFPh:AA 86.67:13.33 mol.%), respectively, using the Š–B method—205.50 kJ/mol (p-PGFPh:AA 6.77:93.23 mol.%) and 203.74 kJ/mol (p-PGFPh:AA 86.67:13.33 mol.%). In both cases, the values are similar, which proves applicability of nonisothermal and isoconversional methods for p-PGFPh:AA copolymers degradation processes analysis. Comparing values obtained for different compositions of p-PGFPh:AA copolymer, it can be said that copolymers with less polyesther resin content (p-PGFPh:AA 6.77:93.23 mol.%) have higher values.

## 4. Conclusions

Data are provided on polypropyleneglycolfumaratephthalate (p-PGFPh) and acrylic acid (AA) based copolymers thermal stability research; those can be used as hydrogels for processing vegetable crops. This is caused by the fact that hydro sorbents can be exposed to high temperature during production and use. It should be noted that the technology for the production of one of the main sources of raw materials—polypropyleneglycolfumaratephthalate, which is necessary for the production of the proposed moisture sorbent, has been fully developed by the authors. It is also known that products of copolymerization of unsaturated polyester resins and vinyl monomers are high-melting and cross-linked copolymers with chaotic grid location in space. As evidence shows, this is related to the fact that radical copolymerization does not allow synthesizing copolymers with required properties completely. Due to this, RAFT-polymerization is of high interest because it allows replacing uncontrolled chain termination reactions with reversible reactions allowing us to control square chain termination. In the follow-up study we suggest using RAFT-polymerization to synthesize polypropyleneglycolfumaratephthalate and acrylic acid based copolymers.

According to preliminary research we have attempted to study kinetic features of p-PGFPh:AA polymers thermal destruction in more details. Kinetic parameters were calculated using Friedman and Ozawa-Flynn-Wall methods as well as non-parametric kinetics (NPK) method. These methods are highly beneficial for kinetic interpretation of thermogravimetric data obtained as result of complex processes such as p-PGFPh:AA copolymers thermal decomposition and can be used without knowledge of decomposition process reaction order. It should be noted that copolymers’ thermal stability decreases in the following order: p-PGFPh:AA (6.77:93.23) > p-PGFPh:AA (86.67:13.33) > p-PGFPh:AA (44.17:55.17). Thus, having summarized experimental data on researching thermal stability, we can assume that p-PGFPh:AA (6.77:93.23; 44.17:55.17 and 86.67:13.33) copolymers are relatively highly resistant to heating. It was also found that activation energy values calculated using FR and FWO methods are in line with calculations using non-parametric kinetics method. Therefore, according to ICTAC recommendations obtained kinetic parameters can be used to describe and optimize reaction conditions for obtaining p-PGFPh:AA copolymers.

Also, due to potential efficiency of the results to be obtained in the future, we can talk about need for further larger research in this area.

## Data Availability

All data presented in this paper are available upon request from the corresponding author.

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
