# Peer review of "Research the Thermal Decomposition Processes of Copolymers Based on Polypropyleneglycolfumaratephthalate with Acrylic Acid"

_polymers, 2023, doi:10.3390/polym15071725_

Round 1
Reviewer 1 Report
Revision for Polymers (ISSN 2073-4360)
Manuscript ID: polymers-2284989
Title: Research the Thermal Decomposition Processes of Copolymers Based on Polypropylenegly-colfumaratephthalate with Acrylic Acid
List of authors: Akmaral Sarsenbekova, Gaziza Zhumanazarova*, Yerkeblan M. Tazhbayev, Gulshakhar Kudaibergen, Saule Kabieva, Zhannur Issina, Aigul Kaldybayeva, Arailym Mukabylova, Muslim Kilybay
This research paper deals with study of the thermal destruction of copolymers based on polypropyleneglycolfumaratephthalate with acrylic acid and the determination of related kinetic parameters of the destruction. I consider the research work a valuable activity in the field of thermal degradation of polymers. As the subject is interesting, I am willing to recommend major revisions with pending manuscript decision. I will gladly be able to review the modified manuscript once the following points have been fully addressed:
1) Page 1, abstract section – I would suggest to add more quantitative data to the abstract. This will highlight the novelty of the research activity.
2) Page 1, introduction, line 9 – I would suggest to add some sentences reporting more examples of application for polyester resins. The authors should also add a list of advantages and drawbacks related to the use of polyester resins.
3) Page 1, line 15 – The aging aspect is also linked to the recycling of polymer organic materials. The authors should briefly mention it and some sentences should be added to highlight the impact of aging on physicochemical properties of polyester resins. Please provide some references to support that.
4) Page 2, line 20 – The authors are saying that there are also other methods to evaluate the aging. The authors should mention them and report some examples where TA methods have been used because most suitable.
5) Page 2, lines 20-25 – These sentences appear not clear. The authors should better explain this part of the introduction.
6) Page 2, results and discussion, lines 1-2 – I would suggest to summarize the practical benefits again at this point.
7) Page 3, figure 1 – Figure 1 should be revised. The x axis should start with the collected data for the curve and the same for the end. I also think that on the y axis should be written Weight (%).
8) Page 3, lines 5-11 – Are the authors meaning 100 °C and not 100 °C/min. The x axis is only expressed in temperature.
9) Page 3, lines 20-28 – The authors are mentioning many vibration bands, but they should also specify if they are stretching or bending ones.
10) Page 4, line 30-31 – There are some letters reported in red color, please revise.
11) Page 4, lines 35-40 – Have the authors considered the possibility to perform a DIP-MS analysis to better identify the species in the gas phase? However, the authors should report some researches supporting the composition of the gas phase, also highlighting the novelty of the present research.
12) Page 5, lines 49-55 – Please revise the size of the letters at this point, there are letters with different size along the whole text.
13) Page 5, equations (2) and (3) – The authors should mention the meaning for the reader of each symbol in the several formulas.
14) Page 9, at the end of “results and discussion” section – How the gas substances can influence the decomposition process at this point? The authors should clarify these sentences.
15) Page 9, conclusion section – The authors should reduce the conclusions to main highlights and achievements. also, in the end of the conclusion section, the authors should add some directions for future research efforts.

Reviewer 2 Report
The paper describes the decomposition kinetics of copolymers. Overall, the results can be of interest to the community, but the discussion level should be improved to reach the necessary scientific level.
Most of the issues has been covered by recent publication, https://doi.org/10.3390/thermo2040029, i recommend the authors to consult it and revised text. More specifically, it is recommended to expand the range of heating rates, to use the lower sample masses. In reporting of isoconversional results, comparison of methods that are known to be of different accuracy has no meaning. Same for reporting the average activation energy for a complex process in study.
Also, i recommend to consult the most recent ICTAC recommendations for analysis of decomposition kinetics https://doi.org/10.1016/j.tca.2022.179384
The last section of comparison of SB and NPK results should be extended. Why the activation energy is not constant for a method you called SB (usually it is the sort of nonlinear regression and Ea is one of the parameters, hence, is constant)?
Additionally, the section with mechanistic findings of the study should also be included. what implications for the recation mechanism are suggested?
Round 2
Reviewer 1 Report
Revision for Polymers (ISSN 2073-4360)
Manuscript ID: polymers-2284989
Title: Research the Thermal Decomposition Processes of Copolymers Based on Polypropylenegly-colfumaratephthalate with Acrylic Acid
List of authors: Akmaral Sarsenbekova, Gaziza Zhumanazarova*, Yerkeblan M. Tazhbayev, Gulshakhar Kudaibergen, Saule Kabieva, Zhannur Issina, Aigul Kaldybayeva, Arailym Mukabylova, Muslim Kilybay
The authors have largely addressed all the comments and question that I raised in my previous review. In particular, the description of the different sections and systems, including some exemplificative figures, the supporting references and the text concerning results already present in the literature are much more improved in the revised manuscript. Overall, I think this is a commendable advancement and therefore I recommend the manuscript for publication on Polymers.

Reviewer 2 Report
Null